# Implementation research on noncommunicable disease prevention and control interventions in low- and middle-income countries: A systematic review

Celestin Hategeka[1]*, Prince Adu[2], Allissa Desloge[3], Robert Marten[4], Ruitai Shao[5], Maoyi Tian[6,7], Ting Wei[6], Margaret E. Kruk[1]

1 Department of Global Health and Population, Harvard TH Chan School of Public Health, Boston, Massachusetts, United States of America, 2 School of Population and Public Health, University of British Columbia, Vancouver, British Columbia, Canada, 3 School of Public Health, University of Illinois Chicago, Chicago, Illinois, United States of America, 4 Alliance for Health Policy and Systems Research, WHO, Geneva, Switzerland, 5 Department of NCD, WHO, Geneva, Switzerland, 6 The George Institute for Global Health, Faculty of Medicine and Health, University of New South Wales, Sydney, Australia, 7 School of Public Health, Harbin Medical University, Harbin, China

* celestin.hategeka@alumni.ubc.ca

## Abstract

### Background

While the evidence for the clinical effectiveness of most noncommunicable disease (NCD) prevention and treatment interventions is well established, care delivery models and means of scaling these up in a variety of resource-constrained health systems are not. The objective of this review was to synthesize evidence on the current state of implementation research on priority NCD prevention and control interventions provided by health systems in low- and middle-income countries (LMICs).

### Methods and findings

On January 20, 2021, we searched MEDLINE and EMBASE databases from 1990 through 2020 to identify implementation research studies that focused on the World Health Organization (WHO) priority NCD prevention and control interventions targeting cardiovascular disease, cancer, diabetes, and chronic respiratory disease and provided within health systems in LMICs. Any empirical and peer-reviewed studies that focused on these interventions and reported implementation outcomes were eligible for inclusion. Given the focus on this review and the heterogeneity in aims and methodologies of included studies, risk of bias assessment to understand how effect size may have been compromised by bias is not applicable. We instead commented on the distribution of research designs and discussed about stronger/weaker designs. We synthesized extracted data using descriptive statistics and following the review protocol registered in PROSPERO (CRD42021252969). Of 9,683 potential studies and 7,419 unique records screened for inclusion, 222 eligible studies evaluated 265 priority NCD prevention and control interventions implemented in 62 countries (6% in low-

**Data Availability Statement:** All relevant data are within the manuscript and its supporting information files.

**Funding:** The Alliance for Policy and Health Systems Research and The World Health Organization funded the study. CH received support through a Banting Postdoctoral Fellowship from the Canadian Institutes of Health Research. The funders had no role in study design, data collection and analysis, decision to publish, or preparation of the manuscript.

**Competing interests:** I have read the journal's policy and the authors of this manuscript have the following competing interests: CH, PA and MEK report personal fees from the funders, during the conduct of the study. MEK is an Academic Editor on PLOS Medicine's editorial board. RM and RS are employed by the study funders.

**Abbreviations:** COVID-19, Coronavirus Disease 2019; LIC, low-income country; LMIC, low- and middle-income country; NCD, noncommunicable disease; SDG, Sustainable Development Goal; UMIC, upper middle-income country.

income countries and 90% in middle-income countries). The number of studies published has been increasing over time. Nearly 40% of all the studies were on cervical cancer. With regards to intervention type, screening accounted for 49%, treatment for 39%, while prevention for 12% (with 80% of the latter focusing on prevention of the NCD behavior risk factors). Feasibility (38%) was the most studied implementation outcome followed by adoption (23%); few studies addressed sustainability. The implementation strategies were not specified well enough. Most studies used quantitative methods (86%). The weakest study design, preexperimental, and the strongest study design, experimental, were respectively employed in 25% and 24% of included studies. Approximately 72% of studies reported funding, with international funding being the predominant source. The majority of studies were proof of concept or pilot (88%) and targeted the micro level of health system (79%). Less than 5% of studies report using implementation research framework.

## Conclusions

Despite growth in implementation research on NCDs in LMICs, we found major gaps in the science. Future studies should prioritize implementation at scale, target higher levels health systems (meso and macro levels), and test sustainability of NCD programs. They should employ designs with stronger internal validity, be more conceptually driven, and use mixed methods to understand mechanisms. To maximize impact of the research under limited resources, adding implementation science outcomes to effectiveness research and regional collaborations are promising.

## Author summary

### Why was the study done?

- While the evidence for the clinical effectiveness of most noncommunicable disease (NCD) prevention and treatment interventions is well established, care delivery models and means of scaling these up to entire populations in need in heterogeneous and resource-constrained health systems are not.

- Implementation research on NCD program delivery can illuminate what does and does not work in preventing NCDs or achieving NCD control. This can promote faster, more efficient, and more effective scale-up of life-saving and health-preserving health system strategies.

- Evidence needed on the current state of implementation research on World Health Organization (WHO) priority NCD prevention and control interventions to help inform research priority.

### What did the researchers do and find?

- We performed a systematic review search in MEDLINE and EMBASE databases from 1990 through 2020 to identify implementation research studies that focused on the WHO priority NCD prevention and control interventions targeting cardiovascular

disease, cancer, diabetes, and chronic respiratory disease and provided within health systems in low- and middle-income countries (LMICs).

- We identified 222 eligible studies that evaluated 265 priority NCD prevention and control interventions implemented in 62 countries (6% in low-income countries and 90% in middle-income countries). The number of studies published has been increasing over time. The majority of interventions were focused on either screening (49%) or treatment (39%), while prevention accounted for only 12%.

- Reviewed studies emphasized a few health areas, such as cervical cancer, with many other high-burden conditions little researched. The majority of studies were proof of concept or pilot, quantitative using weaker study designs and targeted the micro level of health system.

## What do these findings mean?

- While implementation research on priority NCDs has grown substantially, from under 10 studies per year in early 2000s to 51 studies in 2020, this is still vastly incommensurate with the health burden of NCDs.

- Future studies should prioritize implementation at scale, target higher levels health systems (meso and macro levels), and test sustainability of NCD programs. They should employ designs with stronger internal validity, be more conceptually driven, and use mixed methods to understand mechanisms.

- To maximize impact of the research under limited resources, adding implementation science outcomes to effectiveness research and regional collaborations are promising.

## Introduction

Noncommunicable diseases (NCDs) have become the leading contributors to morbidity and mortality worldwide. They are now responsible for 74% of all global deaths, 77% of which occur in low- and middle-income countries (LMICs) [1,2]. Approximately 85% of NCD deaths among people aged 30 and 69 years occur in LMICs [1]. Cardiovascular diseases are the leading causes of NCD mortality, followed by cancers, respiratory diseases, and diabetes [1]. Together, these 4 NCDs are responsible of over 80% of all premature NCD deaths [1]. Risk factors such as tobacco and alcohol use, physical inactivity, and unhealthy diets result in significantly greater risk of dying from NCDs. Primary, secondary, and tertiary prevention strategies are vital in addressing NCD burden [1]. Sustainable Development Goal (SDG) target 3.4 commits countries to reduce premature mortality from NCDs by a third by 2030 relative to 2015 levels. Recent analysis shows that no LMIC is on track to meet this target for both men and women if they maintain their 2010 to 2016 average rates of decline [3].

NCD prevention and control should not be regarded as a vertical issue separated from other health conditions. The ongoing Coronavirus Disease 2019 (COVID-19) pandemic has put a spotlight on NCDs, as these increased the risk of death for people with COVID infection. Similarly, NCDs increase mortality risk among people with other infectious diseases such as tuberculosis and HIV. It further highlighted the economic and social inequities in who is afflicted with NCDs, in both high-income countries and LMICs. While primary prevention relies on public health, taxation, and other public policy measures, mitigating the health

consequences of NCDs also requires strong health systems. Health systems that recognize this challenge and address modifiable risk factors and prioritize the management of NCDs will be better positioned to promote and maintain health. Data from the 2019 World Health Organization (WHO) NCD Country Capacity surveys reveal that only half of 160 countries have national guidelines for NCDs, half have the 6 essential technologies for early detection, diagnosis, and monitoring of NCDs available in primary care facilities of the public health sector, and 20% of countries have 6 (or fewer) of the 11 essential medicines available [4]. Greater prioritization of NCDs within health systems and high-quality care are essential to achieving SDG 3.4 [3]. Beyond this lies an important agenda for tackling the cumulatively large group of rarer NCDs that afflict the world's poorest people [5].

To support countries in crafting effective NCD strategies, the WHO Assembly endorsed the Global Action Plan for the Prevention and Control of Noncommunicable Diseases 2013–2020 (GAP-NCD) in May 2013 together with a set of evidence-based interventions (best-buys) and policy options in its appendix 3 that was updated in 2016 and provides 84 interventions or policy options [6,7]. Furthermore, WHO has developed a compendium including all available health interventions. The list and compendium aim to assist Member States, as appropriate in specific national contexts, in implementing measures to achieve the 9 global voluntary targets for NCDs and Target 3.4 of the SDGs. Despite recent calls for a new commitment to implementation research for NCDs, a mid-point evaluation of the WHO NCD Global Action Plan 2013–2030 (NCD-GAP) found that "research has been the weakest NCD-GAP objective in terms of implementation and that progress in implementing research linked to the NCD-GAP has been slow and incremental" [8,9].

While the evidence for the clinical effectiveness of most NCD prevention and treatment interventions is well established, care delivery models and means of scaling these up to entire populations in need in heterogeneous and resource-constrained health systems are not. Implementation research on NCD program delivery, including cost effectiveness in various regions, can illuminate what does and does not work in achieving NCD control [8,10–12]. This can promote faster, more efficient, and more effective scale-up of life-saving and health-preserving health system strategies [13,14]. In this systematic review, we aim to synthesize evidence on the current state of implementation research on WHO priority NCD prevention and control interventions provided within health systems in LMICs [6,7,15–17].

## Methods

This systematic review was conducted according to a study protocol registered in PROSPERO (#CRD42021252969) [18].

### Search strategy

Following the Systematic Reviews and Meta-Analyses (PRISMA) checklist [19], we searched for implementation research studies that focused on relevant NCD prevention and control interventions (Table A in S1 Appendix) provided within health systems in LMICs and were published in peer-reviewed journals indexed in MEDLINE and EMBASE databases from 1990 to 2020. The databases were last searched on January 20, 2021. Our search terms included medical subject heading (MeSH) terms and/or key words for 4 key themes (implementation research; NCDs; NCD interventions; LMICs) that were adjusted for each database:

- Implementation research (e.g., implementation research, implementation science, diffusion of innovations, implementation strategies, dissemination science, implementation outcomes).

- NCDs (e.g., cardiovascular disease, cancer, diabetes, chronic respiratory disease).

- Interventions (e.g., smoking cessation, management of hypertension, treatment of acute myocardial infarction, cervical and colorectal cancer screening).

- LMICs as defined by the World Bank in 2019 (Table C in S1 Appendix).

Language restrictions were not applied. Full details of the search strategy are provided in Table B in S1 Appendix.

## Inclusion and exclusion criteria

Table 1 summarizes our review's specific eligibility criteria. This review includes peer-reviewed, empirical quantitative, qualitative, and mixed method study designs conducted in

**Table 1. Inclusion and exclusion criteria.**

| | Inclusion criteria | Exclusion criteria |
|---|---|---|
| Population | Human beings with or without NCDs. Human beings with or without NCD risk factors. | Subjects are not human beings. |
| Intervention | NCD prevention and/or control interventions that are provided within health systems (see Table A in S1 Appendix). | Interventions that are not specified in the inclusion criteria. |
| Outcome | Implementation outcomes as defined by Proctor and colleagues and Glasgow and colleagues [20,21]<br>• Acceptability<br>• Adoption<br>• Appropriateness<br>• Feasibility<br>• Fidelity<br>• Penetration<br>• Sustainability<br>• Implementation costs<br>• Reach<br>• Implementation<br>• Maintenance | Outcomes other than those specified in the inclusion criteria. |
| Study design | Quantitative, qualitative, or mixed method.<br>Quantitative study designs included experimental and observational.<br>• Experimental designs:<br>○ Randomized controlled trial,<br>○ Cluster randomized trial,<br>○ Randomized step wedge,<br>• Observational designs:<br>○ Quasi-experimental designs:<br>▪ Single interrupted time series,<br>▪ Controlled interrupted time series,<br>▪ Pre-post with comparison group,<br>▪ Regression discontinuity,<br>▪ Nonrandomized stepped wedge<br>○ Preexperimental designs (no control group or no repeated measures):<br>▪ Pre-post<br>▪ Post-only design<br>○ Other observational designs include:<br>▪ Cohort studies<br>▪ Cross-sectional studies<br>▪ Case-control studies | Nonempirical/primary research including:<br>• Review<br>• Meta-analysis<br>• Editorial<br>• Commentary<br>• Letter to editor<br>• Opinion paper<br>• Newspaper<br>• Protocols<br>• Case report<br>• Epidemiological/descriptive studies (e.g., nonintervention association studies including knowledge, attitude, discrete choice experiment, awareness, willingness, and perception (including perceived barriers) studies) and not in the context of implementation of NCD interventions.<br>• Instrument/screening or diagnostic test validation studies<br>• Call to action<br>• Sharing experience/lessons learned on the field if not resulting from research<br>• (Descriptive) cost-effectiveness studies based on modeling (and not in the context of implementation of NCD interventions) |
| Geographic Scope | LMICs (see Table C in S1 Appendix) | Areas other than LMICs |
| Time frame | 1990–2020 | Studies published before 1990 |

LMIC, low- and middle-income country; NCD, noncommunicable disease.

LMICs that described the implementation of relevant NCD preventive and/or control interventions provided within health systems. Using the updated Appendix 3 of the WHO Global NCD Action Plan 2013–2020, we identified the WHO priority NCD prevention and control interventions [6]. Of these interventions, we selected those that are specifically provided by health systems. This was achieved through discussions and consensus. Table 2 summarizes the intervention categories across eligible NCD risk factors (i.e., tobacco and alcohol use, physical

**Table 2. Summary of eligible NCD preventive and control interventions.**

| Conditions | Intervention categories |
|---|---|
| **NCD risk factors** | |
| Tobacco use | Individual smoking cessation |
| | Mass media campaign smoking cessation |
| Harmful use of alcohol | Alcohol reduction counseling for at risk individuals |
| | Treatment for alcohol use disorder |
| Unhealthy diet | Mass media or other behavior change program to reduce salt intake |
| | Nutrition education in institutions |
| | Salt reduction public institutions |
| | Interventions to promote exclusive breastfeeding |
| Physical inactivity | Community environmental program increase physical activity |
| | Mass media campaign promote physical activity |
| | Physical activity counseling |
| **NCDs** | |
| Cardiovascular disease | Treatment of hypertension |
| | Rehabilitation of post-acute CVD event (myocardial infarction, stroke) |
| | Treatment of high-risk CVD event |
| | Treatment of acute ischemic stroke |
| | Treatment of acute myocardial infarction |
| | Treatment of heart failure |
| | Antibiotic treatment of streptococcal pharyngitis (rheumatic fever prevention) |
| | Treatment for secondary prevention of stroke (e.g., anticoagulation for atrial fibrillation, aspirin) |
| Diabetes | Glycemic control among people with diabetes |
| | Screening to prevent complications among people with diabetes |
| | Treatment of diabetes |
| | Preconception care for women with diabetes |
| | Influenza vaccination for people with diabetes |
| Cancer | Breast cancer screening |
| | Cervical cancer screening |
| | HPV vaccination for teen girls |
| | Colorectal cancer screening |
| | Treatment of breast and colorectal cancer |
| | Hepatitis B immunization for liver cancer prevention |
| | Screening for oral cancer in high-risk groups |
| Chronic respiratory disease | Treatment of asthma and COPD |
| | Influenza vaccination for patients with COPD |

COPD, chronic obstructive pulmonary disease; CVD, cardiovascular disease; HPV, human papilloma virus; NCD, noncommunicable disease.

inactivity, and unhealthy diets) and NCDs (i.e., cardiovascular disease, diabetes, cancer, and chronic respiratory disease), and full details are provided in Table A in S1 Appendix. While our search in databases was not restricted to any language, during study screening/review processes, we only retained eligible studies that were in 6 official languages of the United Nations (i.e., Arabic, Chinese, English, French, Russian, and Spanish). We drew on Proctor and colleagues and Glasgow and colleagues to define implementation outcomes eligible for inclusion [20,21]. Nonempirical/primary research studies are not eligible for inclusion (Table 1).

## Data extraction and analysis

The titles and abstracts of unique results from the databases were reviewed independently by 2 researchers for potential inclusion using COVIDENCE review software [22]. The full texts of studies retained at the title and abstract screening stage were retrieved and independently assessed for inclusion. Any discrepancies were resolved through discussion and consensus. Data extraction on each included study was conducted by a single researcher using a data extraction tool, developed and piloted a priori (Table D in S1 Appendix). Data elements included study characteristics (e.g., publication year, country of implementation, study funding), NCD conditions (risk factors and disease), intervention details (e.g., type of intervention, level of health system), methods (e.g., research approach, study design), implementation outcomes (e.g., fidelity, feasibility), and equity lens (e.g., disaggregated by key SES stratifiers, targeted vulnerable population). We also extracted data on implementation strategies including actor (i.e., who delivered the intervention), action target, and recipients; details of other implementation strategies were not sufficiently described to permit extraction [23]. The recipients of the action/strategy were further aggregated by demographic subgroup (e.g., people eligible for cancer screening including cervical and colorectal), disease risk subgroup (e.g., patients with myocardial infarction, patients with diabetes or hypertension, people who smoke), general population, healthcare workers (e.g., physicians, nurses, pharmacists, and midwives), and community health workers. We synthesize extracted data using descriptive statistics and following the review protocol registered in PROSPERO. Specifically, we provide an overview of NCD priority intervention implementation study characteristics across NCD conditions to shed light on the current state of implementation research of priority NCD prevention and control interventions in LMICs. Given this review does not focus on effect size of NCD interventions, we did not perform a meta-analysis.

## Risk of bias assessment

This review focuses on implementation of multiple interventions across various NCDs, rather than effectiveness of any single set of interventions. Further, studies with heterogenous aims and methodologies (including qualitative methodology) were included. Therefore, risk of bias assessment to understand how effect size may have been compromised by bias is not applicable in this review. We instead commented on the distribution of research designs and discussed about stronger/weaker designs.

## Results

Our search strategy implemented in MEDLINE and EMBASE identified 9,683 publications, of which 7,419 unique records were screened for inclusion. Abstract and full-text screening identified 222 studies that met our inclusion criteria (Tables 1 and 2) [24–245]. A summary of this process is presented in the PRISMA flow diagram in Fig 1.

The 222 studies included in this review evaluated 265 priority NCD prevention and control interventions implemented in 62 countries, of which 6% were in low-income countries (LICs),

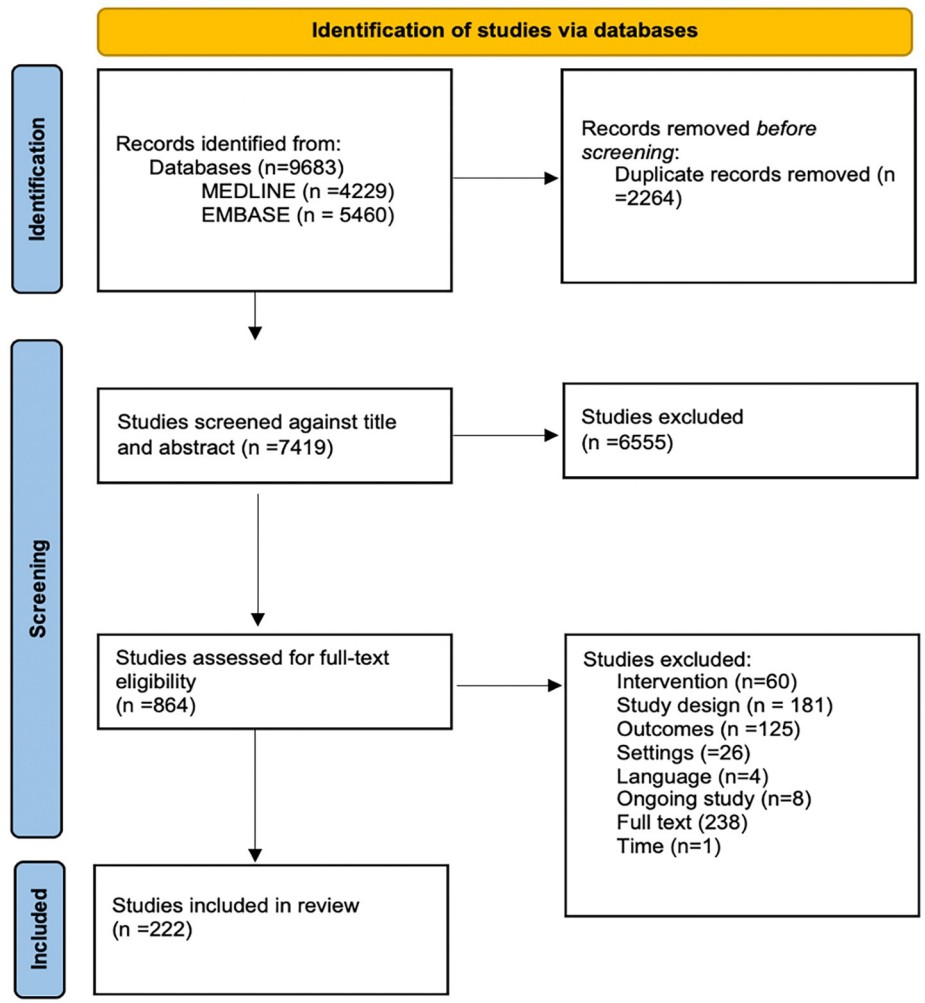

**Fig 1. PRISMA flow chart.** Intervention refers to studies excluded because they studied the implementation of interventions that did not meet the eligible criteria. Study design refers to studies excluded because they used study designs that did not meet eligibility criteria (e.g., nonempirical studies including reviews and commentaries). Outcomes refer to studies excluded based on not having focused on relevant implementation outcomes. Settings refer to studies excluded because they were not conducted in LMICs. Full text means that studies were excluded because full text was not available. Time refers to studies that were excluded because they were published before/conducted before 1990.

45% in LMICs, and 46% in upper middle-income countries (UMICs) (Table 3 and Figs 2, 3, and 4A and Table E in S1 Appendix). The NCD conditions targeted varied by income groups of countries (Fig A in S1 Appendix). Eight of the included studies were multicountry studies. The number of studies published has been increasing over time (Fig 5A). Overall, the majority of interventions were focused on either screening (49%) or treatment (39%), while prevention accounted for only 12%, with nearly 80% of these tackling prevention of the shared NCD behavioral risk factors—tobacco use, unhealthy diet, physical inactivity, and harmful use of alcohol. The NCD interventions varied by conditions and type (prevention, screening, and treatment) (Figs 2, B, and C in S1 Appendix). Notably, over one-third of the interventions studied (37%) were for cervical cancer (Fig 2), which accounts for 0.35% of DALYs lost and

**Table 3. Overview of study characteristics.**

| NCDs and risk factors | Intervention categories | N | Distribution of priority NCD interventions (N = 265) | | | | | | | Implementation strategies | | |
|---|---|---|---|---|---|---|---|---|---|---|---|---|
| | | | Country's income classification, N | Methods approach, N | Major study design, N | Health system level*, N | Level of scale-up, N | Implementation outcomes, N | Considered equity†, N | Actor, N | Action target, N | Recipients, N |
| Tobacco use | Individual smoking cessation | 6 | LMICs = 5, UMICs = 1 | Quantitative = 5, Mixed = 1 | Experimental = 2, Multiple = 1, Preexperimental = 2, Other Observational = 1 | Micro = 4, Meso = 2 | Pilot† = 5, Scale-up = 1 | Adoption = 1, Appropriateness = 1, Feasibility = 3, Multiple = 1 | 4 | Researchers = 4, Providers = 2 | Behavior = 6 | Disease risk subgroup = 6 |
| | Mass media campaign smoking cessation | 2 | UMICs = 2 | Quantitative = 2 | Experimental = 1, Observational = 1 | Macro = 2 | Scale-up = 2 | Adoption = 1, Penetration = 1 | 1 | Researchers = 1, MOH = 1 | Behavior = 2 | Disease risk subgroup = 2 |
| Harmful use of alcohol | Alcohol reduction | 1 | LMICs = 1 | Quantitative = 1 | Experimental = 1 | Micro = 1 | Pilot = 1 | Multiple = 1 | 0 | Researchers = 1 | Behavior = 1 | Disease risk subgroup = 1 |
| Unhealthy diet | Mass media or other behavior change program to reduce salt intake | 3 | LMICs = 2, UMICs = 1 | Quantitative = 3 | Experimental = 2, Other observational = 1 | Micro = 1, Meso = 1, Macro = 1 | Pilot = 1, Scale-up = 2 | Adoption = 1, Penetration = 1, Multiple = 1 | 2 | Researchers = 1, MOH = 2 | Behavior = 3 | General population = 2, Disease risk subgroup = 1 |
| | Nutrition education in institutions | 5 | LMICs = 1, UMICs = 3, Multiple = 1 | Quantitative = 4, Mixed method = 1 | Quasi-experimental designs = 3, Preexperimental = 1, Other observational = 1 | Micro = 1, Meso = 3, Macro = 1 | Pilot = 2, Scale-up = 3 | Acceptability = 1, Adoption = 2, Feasibility = 1, Penetration = 1 | 3 | Researchers = 4, MOH = 1 | Behavior = 3, Behavior, health outcomes = 2 | Demographic subgroup = 2, Disease risk subgroup = 3 |
| | Salt reduction public institutions | 2 | UMICs = 2 | Quantitative = 2 | Other observational = 2 | Macro = 2 | Pilot = 1, Scale-up = 1 | Adoption = 1, Penetration = 1 | 1 | Researchers = 1, MOH = 1 | Behavior = 2 | Demographic subgroup = 2 |
| Physical inactivity | Community environmental program increase physical activity | 4 | LMICs = 3, UMICs = 1 | Quantitative = 4 | Experimental = 2, Preexperimental = 1, Other observational = 1 | Micro = 1, Meso = 1, Macro = 2 | Pilot = 1, Scale-up = 3 | Feasibility = 1, Penetration = 1, Multiple = 2 | 2 | Researchers = 3, MOH = 1 | Behavior = 2, Behavior and knowledge = 2 | Demographic subgroup = 3, Disease risk subgroup = 1 |
| | Mass media campaign promote physical activity | 2 | UMCIs = 2 | Quantitative = 2 | Experimental = 1, Other observational = 1 | Macro = 2 | Scale-up = 2 | Adoption = 1, Penetration = 1 | 1 | Researchers = 1, MOH = 1 | Behavior = 1, Behavior and knowledge = 1 | Demographic subgroup = 1, Disease risk subgroup = 1 |
| CVD | Rehabilitation post-acute CVD event | 1 | UMICs = 1 | Quantitative = 1 | Experimental = 1 | Micro = 1 | Pilot = 1 | Feasibility = 1 | 0 | Researchers = 1 | Health outcomes = 1 | Disease risk subgroup = 1 |
| | Treatment of high-risk CVD event | 5 | LMICs = 2, UMICs = 3 | Quantitative = 5 | Experimental = 2, Quasi-experimental designs = 2, Other observational = 1 | Micro = 5 | Pilot = 5 | Acceptability = 1, Adoption = 2, Feasibility = 1, Maintenance = 1 | 3 | Researchers = 4, Providers = 1 | Behavior = 3, Health outcomes = 1 | Demographic subgroup = 1, Disease risk subgroup = 3, HCWs = 1 |
| | Treatment of acute ischemic stroke | 10 | LMICs = 5, UMICs = 5 | Quantitative = 10 | Experimental = 2, Preexperimental = 7, Other observational = 1 | Micro = 6, Meso = 4 | Pilot = 8, Scale-up = 2 | Adoption = 4, Feasibility = 4, Fidelity = 1 | 1 | Researchers = 5, MOH = 4, Providers = 1 | Health outcomes = 10 | Disease risk subgroup = 10 |
| | Treatment of acute myocardial infarction | 12 | LMICs = 2, UMICs = 10 | Quantitative = 11, Qualitative = 1 | Experimental = 3, Quasi-experimental designs = 2, Preexperimental = 4, Other observational = 3 | Micro = 6, Macro = 6 | Pilot = 10, Scale-up = 1 | Adoption = 5, Feasibility = 5, Fidelity = 1, Penetration = 1 | 3 | Researchers = 4, MOH = 4, Providers = 4 | Health outcomes = 11, Behavior = 1 | Disease risk subgroup = 11, HCWs = 1 |

(Continued)

Table 3. (Continued)

| NCDs and risk factors | Intervention categories | N | Country's income classification, N | Methods approach, N | Major study design, N | Health system level*, N | Level of scale-up, N | Implementation outcomes, N | Considered equity†, N | Implementation strategies | | |
|---|---|---|---|---|---|---|---|---|---|---|---|---|
| | | | | | | | | | | Actor, N | Action target, N | Recipients, N |
| | Treatment of heart failure | 5 | LMICs = 2 UMICs = 3 | Quantitative = 5 | Experimental = 2 Quasi-experimental designs = 1 Preexperimental = 2 | Micro = 5 | Pilot = 5 | Adoption = 2 Feasibility = 3 | 1 | Researchers = 3 Providers = 2 | Health outcomes = 5 | Disease risk subgroup = 5 |
| | Treatment of hypertension | 23 | LMICs = 10 Multiple = 2 UMICs = 11 | Quantitative = 16 Qualitative = 1 Mixed method = 6 | Experimental = 7 Quasi-experimental designs = 4 Preexperimental = 2 Other observational = 5 Multiple = 5 | Micro = 20 Meso = 2 Macro = 1 | Pilot = 22 Scale-up = 1 | Adoption = 1 Feasibility = 18 Fidelity = 3 Multiple = 1 | 10 | Researchers = 10 MOH = 5 NGO = 1 Providers = 6 NC = 1 | Behavior = 10 Health outcomes = 8 Behavior and health outcomes = 5 | Demographic subgroup = 1 Disease risk subgroup = 22 |
| Diabetes | Glycemic control among people with diabetes | 7 | LMICs = 3 UMICs = 2 Multiple = 2 | Quantitative = 5 Mixed method = 2 | Experimental = 2 Quasi-experimental designs = 4 Multiple = 1 | Micro = 4 Meso = 2 Macro = 1 | Pilot = 6 Scale-up = 1 | Adoption = 1 Appropriateness = 2 Feasibility = 2 Multiple = 2 | 4 | Researchers = 5 MOH = 1 NGO = 1 | Behavior = 4 Health outcomes = 2 Behavior and health outcomes = 1 | Demographic subgroup = 1 Disease risk subgroup = 6 |
| | Screening to prevent complications among people with diabetes | 17 | LMICs = 10 UMICs = 7 | Quantitative = 16 Mixed method = 1 | Experimental = 1 Quasi-experimental designs = 1 Preexperimental = 6 Other observational = 8 Multiple = 1 | Micro = 16 Meso = 1 | Pilot = 1 | Acceptability = 1 Adoption = 2 Feasibility = 10 Multiple = 1 Reach = 3 | 6 | Researchers = 8 MOH = 1 NGO = 3 Providers = 4 NC = 1 | Behavior = 5 Health outcomes = 12 | Disease risk subgroup = 16 CHWs = 1 |
| | Diabetes management | 39 | LMICs = 22 UMICs = 14 Multiple = 3 | Quantitative = 33 Qualitative = 1 Mixed method = 5 | Experimental = 8 Quasi-experimental designs = 6 Preexperimental = 9 Other observational = 12 Multiple = 4 | Micro = 34 Meso = 3 Macro = 2 | Pilot = 38 Scale-up = 1 | Acceptability Adoption = 1 Appropriateness = 2 Feasibility = 21 Fidelity = 1 Reach = 3 Multiple = 4 | 16 | Researchers = 22 MOH = 6 NGO = 4 Providers = 6 NC = 1 | Behavior = 11 Behavior and knowledge = 3 Health outcomes = 20 Behavior and health outcomes = 4 Knowledge and health outcomes = 1 | Demographic subgroup = 3 Disease risk subgroup = 35 CHWs = 1 |
| | Influenza vaccination for people with diabetes | 1 | UMICs = 1 | Quantitative = 1 | Other observational = 1 | Micro = 1 | Pilot = 1 | Adoption = 1 | 0 | Researchers = 1 | Health outcomes = 1 | Disease risk subgroup = 1 |
| Cancer | Breast cancer screening | 9 | LMICs = 5 UMICs = 4 | Quantitative = 9 | Experimental = 1 Quasi-experimental designs = 2 Preexperimental = 2 Other observational = 4 | Micro = 6 Macro = 9 | Pilot = 7 Scale-up = 1 | Acceptability = 1 Adoption = 1 Feasibility = 4 Implementation cost = 1 Reach = 1 Multiple = 1 | 7 | Researchers = 4 MOH = 4 NC/NA = 1 | Behavior = 4 Health outcomes = 5 | Demographic subgroup = 9 |

(Continued)

**Table 3.** (Continued)

| NCDs and risk factors | Intervention categories | N | Distribution of priority NCD interventions (N = 265) | | | | | | | Implementation strategies | | |
|---|---|---|---|---|---|---|---|---|---|---|---|---|
| | | | Country's income classification, N | Methods approach, N | Major study design, N | Health system level*, N | Level of scale-up, N | Implementation outcomes, N | Considered equity†, N | Actor, N | Action target, N | Recipients, N |
| | Cervical cancer screening | 93 | LICs = 13 LMICs = 42 UMICs = 34 Multiple = 4 | Quantitative = 78 Qualitative = 8 Mixed method = 7 | Experimental = 16 Quasi-experimental designs = 4 Preexperimental = 20 Other observational = 47 Multiple = 6 | Micro = 78 Meso = 10 Macro = 5 | Pilot = 78 Scale-up = 12 | Acceptability = 22 Adoption = 23 Feasibility = 22 Implementation cost = 4 Maintenance = 1 Reach = 10 Sustainability = 2 Multiple = 9 | 40 | Researchers = 56 MOH = 13 NGO = 10 Providers = 2 NC = 6 NA = 6 | Behavior = 30 Health outcomes = 41 Knowledge = 5 Knowledge and behavior = 4 Knowledge, behavior, health outcome = 2 Knowledge and health outcomes = 3 NC/NA = 8 | Demographic subgroup = 80 Disease risk subgroup = 2 HCWs = 3 CHWs = 2 |
| | HPV vaccination for teen girls | 5 | LMICs = 2 UMICs = 2 Multiple = 1 | Quantitative = 4 Qualitative = 1 | Preexperimental = 2 Other observational = 3 | Micro = 3 Macro = 2 | Pilot = 3 Scale-up = 2 | Adoption = 3 Feasibility = 1 Multiple = 1 | 1 | Researchers = 3 MOH = 1 NGO = 1 | Behavior = 2 Health outcomes = 3 | Demographic subgroup = 5 |
| | Colorectal cancer screening | 11 | LMICs = 1 UMICs = 10 | Quantitative = 10 Mixed method = 1 | Experimental = 2 Quasi-experimental designs = 1 Preexperimental = 1 Other observational = 6 Multiple = 1 | Micro = 7 Meso = 4 | Pilot = 9 Scale-up = 2 | Acceptability = 1 Adoption = 4 Feasibility = 3 Implementation cost = 1 Reach = 1 Multiple = 1 | 4 | Researchers = 7 MOH = 4 | Behavior = 5 Health outcomes = 5 Knowledge and behavior = 1 | Demographic subgroup = 10 Disease risk subgroup = 1 |
| Chronic respiratory disease | Treatment of asthma | 2 | LMICs = 2 | Mixed method = 2 | Multiple = 2 | Micro = 1 Macro = 1 | Pilot = 1 Scale-up = 1 | Acceptability = 1 | 1 | Researchers = 1 MOH = 1 | Health outcomes = 2 | Disease risk subgroup = 2 |

*Micro level refers to the point where the care providers interact with the patient; micro-level interventions aim to directly influence the performance of the staff or the operations of a facility [11,264]. Meso level refers to the level responsible for service areas/clinical programs providing care for a similar group of patients, typically part of a larger organization (e.g., subnational intervention targeting improvement of a network of facilities and communities) [11,264]. Macro level is the highest (strategic) level of the system, an umbrella including all intersecting areas, departments, providers, and staff (e.g., boards, healthcare network, integrated health system that includes several organizations); macro-level interventions are best able to directly tackle the social, political, economic, and organizational structures that shape a health system [11,264].

†Equity lens used if studies disaggregated by SES stratifiers (e.g., age, sex, education, income, and rural vs. urban) and/or targeted vulnerable population.

CHW, community health workers include ASHAs in India; CVD, cardiovascular disease; HCW, healthcare worker; HPV, human papilloma virus; LIC, low-income country; LMIC, lower middle-income country; MOH, Ministry of Health/Government; N, number of NCD interventions; NC/NA, not clear/not applicable; NCDs, noncommunicable disease; NGO, nongovernmental organization; UMIC, upper middle-income country.

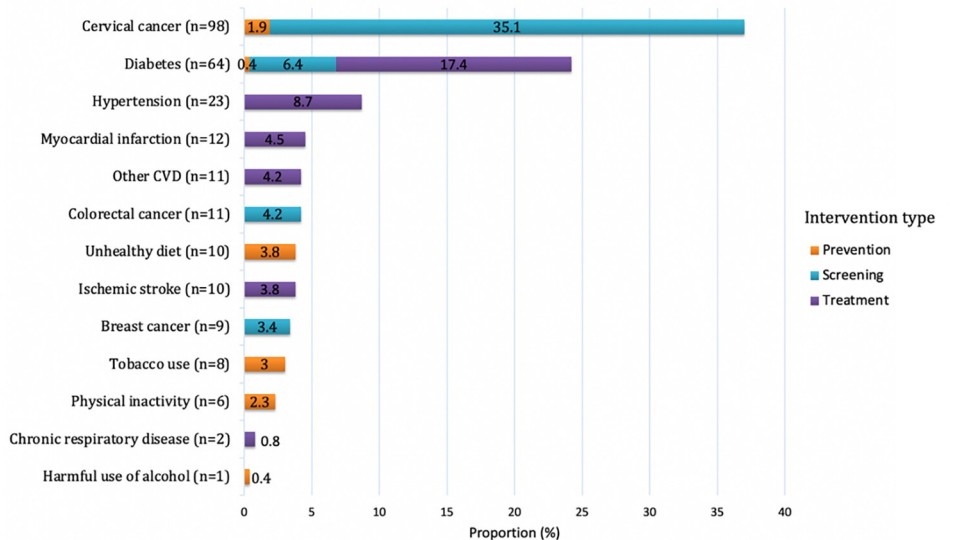

**Fig 2. Distribution of priority NCD prevention and control interventions by type of NCD and their risk factors (*N* = 265).**

0.5% of deaths globally, with similar figures for LMICs (https://vizhub.healthdata.org/gbd-compare/). Diabetes was the focus of nearly one-quarter of the research with hypertension the topic of another 9% (Fig 2). Each of the other recommended interventions represented 5% or

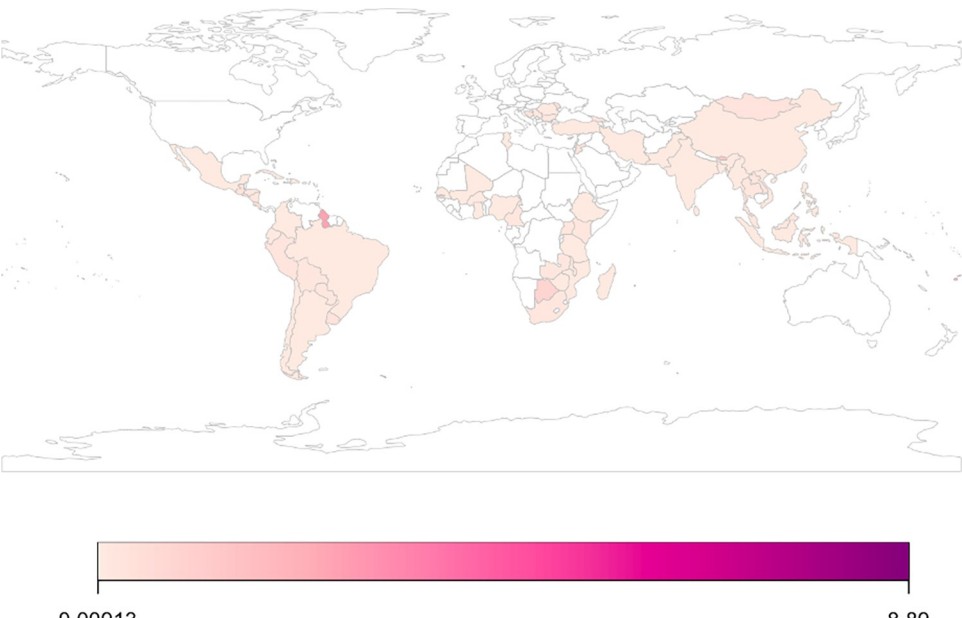

**Fig 3. Distribution of studies per 1 million population by country of implementation.** We used country population size in 2020 (https://data.worldbank.org/indicator/SP.POP.TOTL) to standardized estimates expressed as number of studies per 1 million population. We used "rworldmap" package (https://cran.r-project.org/web/packages/rworldmap/rworldmap.pdf) available in R software to present these standardized estimates across countries where interventions were implemented. Country borders in this package are derived from Natural Earth data. Table E in S1 Appendix shows number of included studies per country.

## A. Countries where interventions were implemented

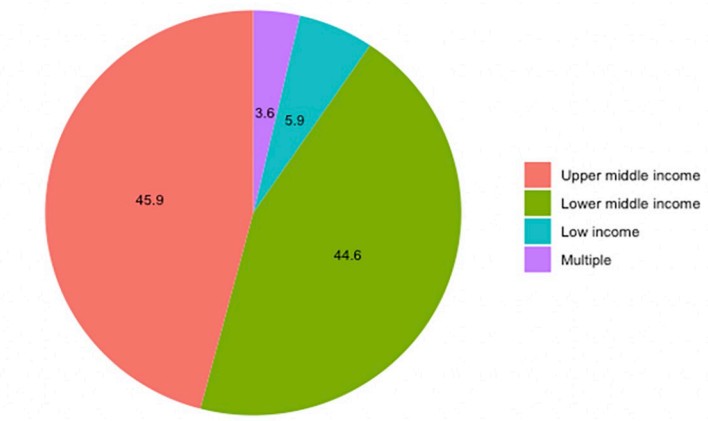

## B. Funding type

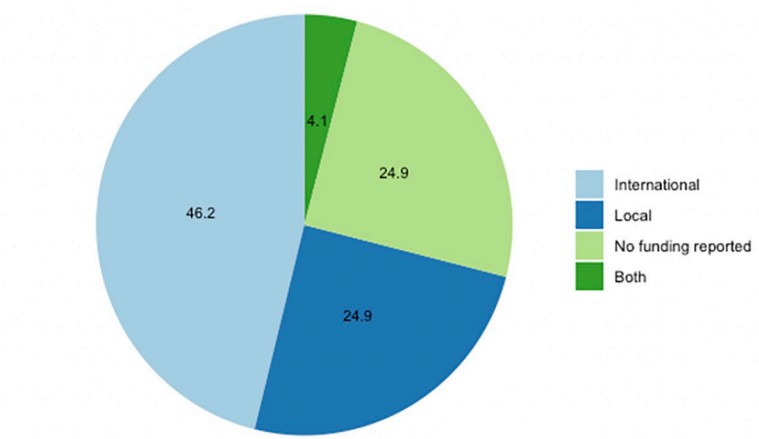

## C. Authorship

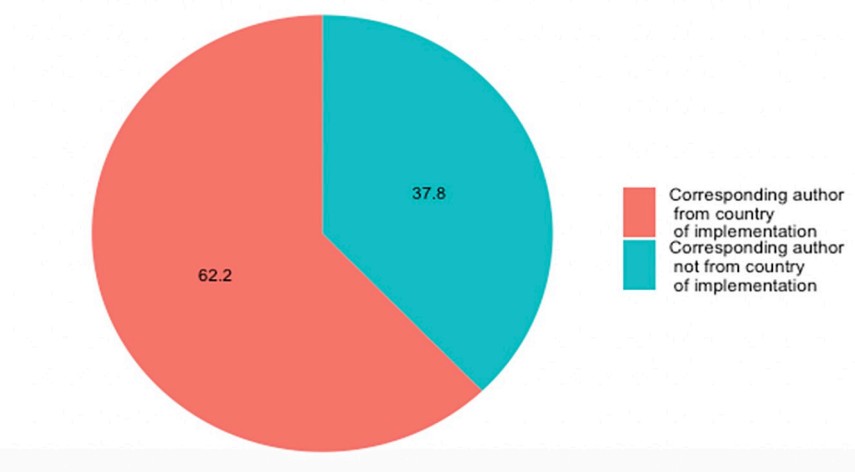

**Fig 4. Distribution of study countries, funding, and authorship (*N* = 222).**

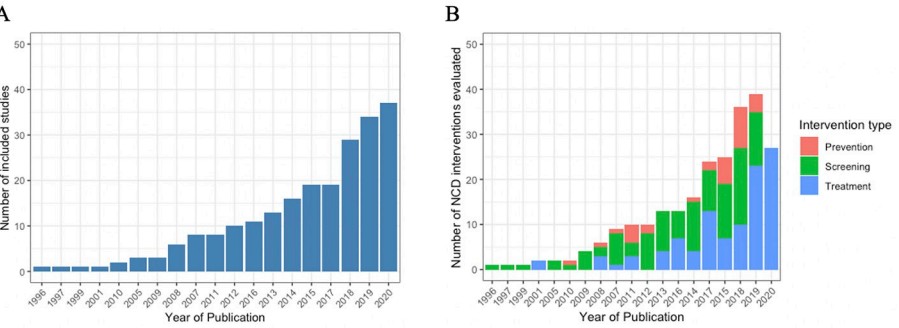

**Fig 5. Growth of research over time (A) and distributions of NCD interventions by type (B).** Fig 5A shows number of studies published each year (*N* = 222 studies); Fig 5B shows distributions by type of interventions (*N* = 265 NCD interventions evaluated in studied included in the review).

less of the implementation research output. Chronic respiratory disease was understudied relative to its prevalence: less than 1% of the studies examined chronic respiratory disease treatment and only 3% smoking cessation programs. The intervention focus appears to vary by income groups of countries (Fig D in S1 Appendix). Feasibility was the most studied implementation outcome followed by adoption (Fig 6). Most of the actors were researchers, which accounted for 58%; whereas government/ministry of health, providers, and NGOs accounted for 18%, 10%, and 6%, respectively. The majority of intervention targeted improvement in health outcomes (45%) followed by change in behavior (34%).

Most studies used quantitative methods, which accounted for 86%, whereas mixed methods and qualitative methods accounted for 9% and 5%, respectively (Table 2). The majority of studies used observational designs, with cross-sectional designs used in 45 studies. Among evaluations, preexperimental studies (such as pre-post without a comparison group or post-only) was the most frequently employed (*n* = 56 or 25% of all studies); experimental designs were used in a quarter of studies (*n* = 53 or 24% of all studies); quasi-experimental evaluation designs (such as pre-post comparison group or time series) were used in 15 papers (7% of all studies) (Fig 7). Study designs also appear to vary by NCD conditions targeted (Fig E in S1 Appendix). The sample size among included studies varied, ranging from 11 to 350,581, with median of 658. Most studies were standalone implementation studies (85%), with some variations by NCD conditions (Fig F in S1 Appendix). Hybrid implementation and effectiveness studies accounted only for 15%. Less than 5% of studies reported they were guided by widely known implementation science framework. Majority of studies were proof of concept or pilot versus scale-up studies (88% versus 12%), with variations by NCD conditions (Fig G in S1 Appendix). The level of health system targeted most often was micro level, accounting for 79% of studies, with variations by NCD conditions. The meso and macro levels of health systems were targeted by 14% and 7% of studies, respectively (Fig H in S1 Appendix). Approximately 42% of studies employed an equity lens—i.e., studies disaggregated by SES stratifiers (e.g., age, sex, education, income, and rural versus urban) and/or targeted vulnerable population.

A majority of studies (72%) reported funding, with international funding being the predominant source (Fig 4B). There seems to be some variations by NCD conditions (Figs I–K in S1 Appendix). For example, while 78% of studies focused on cervical cancer reported funding, of which 77% were from international sources, those focused on colorectal cancer and treatment of acute myocardial infarction received most of their funding from the countries where implementation research was conducted (Fig K in S1 Appendix). Majority of reported funding

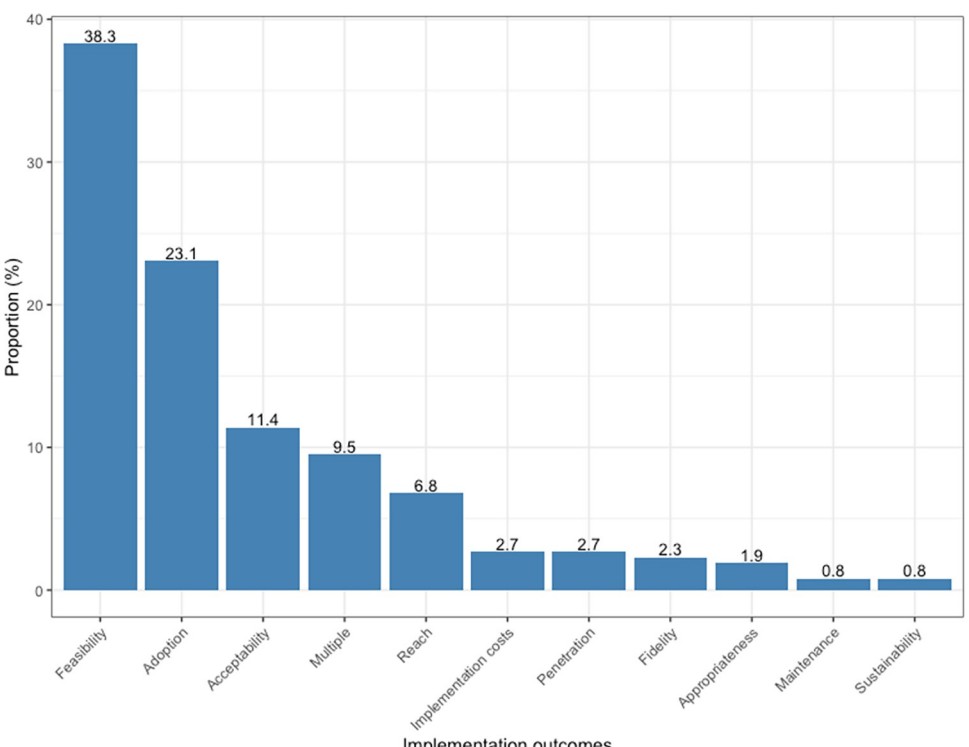

**Fig 6. Distribution of implementation outcomes.**

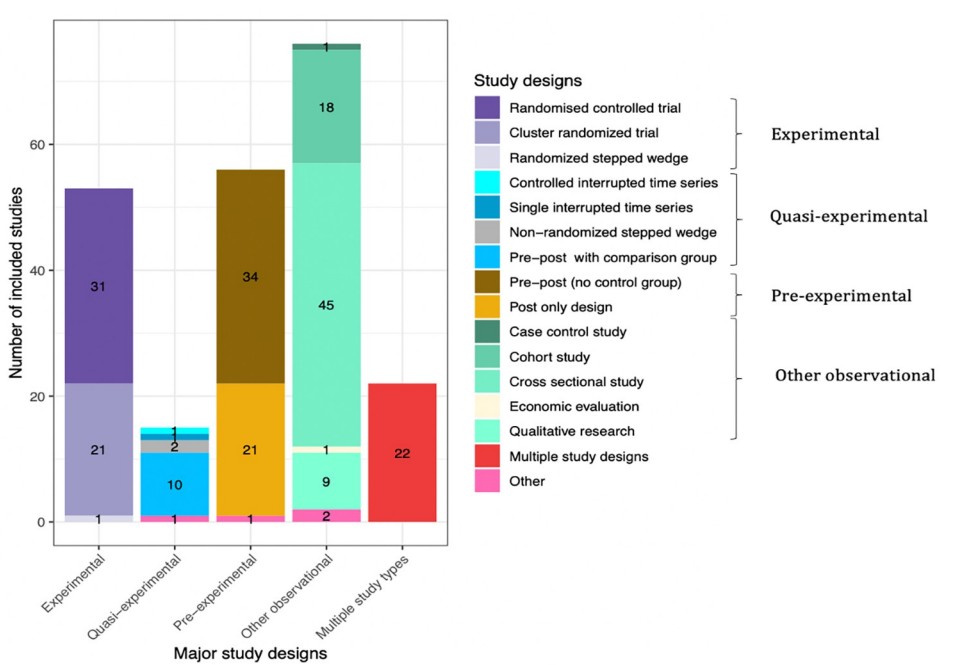

**Fig 7. Study designs.**

was provided by government/universities (43.6%), 35% reported multiple funders, 16% were foundations/NGOs, and 6% were private funders (e.g., pharmaceutical companies, professional associations) (Fig L in S1 Appendix). Approximately 62% of corresponding authors were from the country of implementation (Fig 4C); however, this varied by funding sources, with studies funded by international funders having the highest number of international corresponding authors.

## Discussion

We conducted a systematic review of implementation research studies on NCD prevention and control strategies in LMICs published between 1990 and 2020. We focused our analysis on WHO-recommended NCD interventions carried out by the health system rather than through policy, legislation, or public health approaches [6,7]. These studies therefore represent the state of the implementation science in prevention and control of NCDs by health systems in the countries bearing the bulk of disease burden from noncommunicable conditions.

Of the 222 implementation science studies included in this review, 94% were conducted in middle-income countries (evenly split between lower- and upper-middle) and 6% in LICs. UMICs were slightly overrepresented compared to their share of the LMIC population (approximately 40%). Only 8 of the studies were multicountry studies, suggesting that cross-national generalizability is not the primary motivation for this type of research. India and China, with 43% of the population of LMICs, comprised one-third of the studies. South Africa, Brazil, Iran, Kenya, and Nigeria, were well represented, each contributing more than 3% of the research.

The studies described 265 different NCD interventions, ranging from screening to prevention to treatment and palliation. Conditions studied varied substantially by region. All 13 of the interventions studied in LICs were for cervical cancer screening. In low-middle income countries, cervical cancer accounted for 37%, diabetes for 29%, and hypertension for 8% of interventions. There was a larger variety of conditions studied in UMICs: while cervical cancer and diabetes comprised half the studies, hypertension, myocardial infarction, colorectal cancer, other cardiovascular diseases, and unhealthy diet each comprised more than 5% of studies. The 2 countries with the largest research output and populations, China and India, differed substantially in focus. In India over 70% of studies were on 2 conditions: diabetes (51%) and cervical cancer (19%), whereas the research was more evenly distributed across the NCDs in China.

Half of all studied interventions in this review evaluated screening for disease, nearly 40% treatment and 12% prevention. Over 70% of all screening studies were for cervical cancer, with less research on other conditions for which screening can be cost effective, such as diabetes, colorectal cancer, and breast cancer. Primary and secondary prevention can reduce incidence of disease and forestall disease progression and disability. We found that only 31 (12%) of the studied interventions addressed prevention with nearly 80% of these tackling prevention of the NCD behavior risk factors (e.g., tobacco use, inactivity, unhealthy diet). Less than 10% of the interventions evaluated in this review focused on management of hypertension (the leading metabolic risk factor worldwide, accounting for approximately 19% of global deaths) [246]. This suggests a substantial implementation research gap in secondary prevention, a critical function of primary care and other levels of health systems. Primary care services such as hypertension management and glucose control play a major role in reducing mortality, thus insufficient research on their optimal implementation is a major missed opportunity. Recent work shows that treatment and control rates for hypertension were below 25% and 10%, respectively, in many countries in South Asia and sub-Saharan Africa. These countries also showed the slowest rates of improvement from 1990 [247].

The preponderance of interventions studied was in pilot phase, with fewer than 15% studying large-scale implementation. Along the same lines, feasibility and adoption were the most studied implementation outcomes, suggesting the research is focused on introduction of new approaches. While proof of concept studies is vital with new implementation strategies, arguably WHO-proposed interventions are well established and evidence on (clinical) effectiveness abound. To provide useful guidance to health system planners and realize population health gains, there needs to be a greater investment in large-scale NCD implementation research to promote sustainability of evidence-based interventions. To best scale scarce research resources and accelerate impact, countries could join regional consortia to study interventions and undertake factorial designs that compare locally adapted implementation approaches.

Over three-quarters of the studies were situated at the micro level of the health system—targeting patient, provider, or clinic levels. Nearly 1 in 5 tested a new technology, despite evidence that technology adoption without substantial integration into policies, data, and workflows is typically ineffective in transforming care [248,249]. Education was another common target featuring in 3 of 10 studies; researchers accounted for the majority of the actors. While micro-level approaches are the most "researchable," as they are easiest to implement and analyze; positive results are difficult to scale and sustain in the absence of systemic health system change. The Lancet Commission on High Quality Health Systems notes that high-quality care results from structures that align system aims and policies with strong governance, management, and appropriately trained workforce [250]. In this context, micro-level innovation cannot raise quality system wide and is only effective if undertaken as part of a learning health system that can determine whether it offers sufficient benefit over current practice in complexity, cost, and health benefit, and if so, how to best integrate into the health system [251].

Nearly 9 in 10 studies were stand-alone implementation research. This also points to an opportunity to add implementation research to ongoing effectiveness trials. Integrated or hybrid effectiveness-implementation studies are increasingly being used in high-income countries to shed light on both the outcome and extent and quality of service/program delivery [252]. Notably, fewer than 5% of studies cited use of an implementation science framework consistent with prior research showing that the use of implementation science framework is substantially lower in LMICs compared with high-income countries [253]. The use of a tested conceptual framework can improve the rigor of the research and promote comparability of results. Of the studies that reported a funding source, 60% was from international sources, 33% from the country of the research, and the remaining from both local and international sources. This reflects the low spending for health research and especially for health systems and implementation research in LMICs. The lack of domestic support is unlikely to be offset by global funding going forward; a recent analysis showed that NCDs were under prioritized in bilateral agency portfolios relative to their health impacts [254]. Over 40% of development assistance for health in LICs for NCDs came from NGOs and philanthropies, which are less inclined to support research than operations [254]. Indeed, we found that only 16% of studies with funding information reviewed were supported by philanthropies or NGOs, while the other remaining studies reported funding sources from government, private, and/or multiple sources.

Scarcity of funding for research is a key constraint to needed implementation research for NCDs. While there are proposals for coordinating and increasing global support, it is unrealistic to expect this to meet the scale of needed research without a substantial increase in countries' investment in research [255]. Such an investment is likely to pay off in better health and higher quality, more efficient service delivery [256]. To make best use of research funds, implementation science should strive to be as generalizable as possible—at minimum at a regional level where health systems share similarities. International and regional institutions can play

an important role in supporting research consortia and partnerships to promote efficiency of and accelerate the pace of research and, ultimately its uptake into routine care at scale.

Over 50 of the 222 included studies used an experimental research design. While this is the strongest design to yield causal inference, it is not always feasible to implement. Quasi-experimental designs, such as pretest, posttest comparison group designs, and interrupted time series, which can offer robust information were used in only 15 studies. Preexperimental designs that do not include a comparison group or tracking over time, comprised nearly a quarter of the studies. These designs have very low internal validity and should generally be avoided. The remainder of the studies used cross-sectional descriptions, cohort studies, and qualitative research or multiple study types. Given the disproportionate health harms of NCDs among the poor and other vulnerable groups within countries, disaggregated or stratified analysis is crucial. Forty percent of the assessed studies included stratification by age, sex, education, or urbanicity. Going forward, greater use of quasi-experimental designs, hybrid implementation studies and mixed methods approaches, would benefit the field. An expanded focus on equity of implementation outcomes is also needed.

## Strengths and limitations

Our study had several strengths, notably the extensive scope for the search that covered LMICs, a wide range of outcomes and study types, and a large contingent of conditions and health services. We had no language restrictions permitting a comprehensive assessment of the published literature. The review also had several limitations. We focused on WHO-recommended interventions, which at present do not include guidance for some prevalent conditions such as mental health problems and kidney disease [6,12]. Mental health is a major contributor to the global burden of disease and future work should assess the implementation science for the growing range of mental health interventions that appear to be effective in lower-income settings [257,258]. The studies we assessed used differing definitions of implementation outcomes (e.g., acceptability was measured in some studies by self-report and in others by behavior change). This limits direct comparison of study outcomes. Greater use of implementation science frameworks can promote coherence in the research approaches and terminology used to the benefit of end users. Similarly, given the implementation strategies were not specified well enough in the included studies, we elected to focus on actors, action target, and recipients in our description of implementation strategies. Clearly, reporting empirical implementation studies using existing framework to describe implementation strategies would help bolster uptake of implementation research in NCDs.

We also did not search the gray literature and as such, some relevant studies may have been missed. However, studies in gray literature that were not peer reviewed would have not have been eligible for inclusion in this review. Despite using rigorous search strategies without language restrictions, studies published in journals not indexed in MEDLINE and EMBASE were not captured [259–263]. Given the focus on this review and the heterogeneity in aims and methodologies of included studies, risk of bias assessment to understand how effect size may have been compromised by bias is not applicable. As such, we only commented on the distribution of research designs and discussed about stronger/weaker designs. Lastly, we reported year of publication and not time of when study/implementation was conducted.

## Conclusions

High-quality implementation science can play a key role in informing effective delivery of health system interventions to mitigate the burden of NCDs and avoiding expensive mistakes. While implementation research on priority NCDs has grown substantially, from under 10

studies per year in early 2000s to 51 studies in 2020, this is still vastly incommensurate with the health importance of the topic. Further, the concentration of studies in a few geographies and a few health areas, such as cervical cancer, highlights the dearth of research for other key conditions. We found a major gap in research on secondary prevention, i.e., management of risk factors or early disease to prevent disease progression and premature death. Research on ways in which health systems can be strengthened, including primary care levels, to provide optimal care for NCDs is critically needed. Future studies should use implementation science frameworks, and, when testing interventions, strong research designs with strong internal validity, including well-designed quasi-experimental studies. Opportunities exist for adding implementation science studies to planned effectiveness research.

## Supporting information

**S1 PRISMA Checklist. PRISMA 2020 checklist.**
(DOCX)

**S1 Appendix. Appendix tables and figures.** Table A in S1 Appendix. Interventions provided within health systems. Table B in S1 Appendix. Sample of the search strategy used in the MEDLINE database. Table C in S1 Appendix. List of low- and middle-income countries. Table D in S1 Appendix. Data extraction tool. Table E in S1 Appendix. Distribution of studies by countries where they were implemented. Fig A in S1 Appendix. Variation of conditions evaluated by income group. Fig B in S1 Appendix. Priority NCD interventions ($n = 265$) identified in 222 studies included in the review. Fig C in S1 Appendix. Distribution of included studies by NCD. Fig D in S1 Appendix. Distribution of intervention type by income group. Fig E in S1 Appendix. Distributions by research designs. Fig F in S1 Appendix. Distributions by stand-alone implementation studies vs. embedded or hybrid effectiveness-implementation studies. Fig G in S1 Appendix. Distributions by pilot vs. scale-up project. Fig H in S1 Appendix. Variation by level of health system. Fig I in S1 Appendix. Studies that reported funding (vs. those that did not) by NCD conditions. Fig J in S1 Appendix. Distributions by funding type. Fig K in S1 Appendix. Distribution of funding sources by NCDs and their risk factors. Fig L in S1 Appendix. Types of reported funding sources ($N = 222$ included studies).
(DOCX)

## Author Contributions

**Conceptualization:** Celestin Hategeka, Robert Marten, Ruitai Shao, Margaret E. Kruk.

**Data curation:** Celestin Hategeka, Prince Adu, Allissa Desloge, Maoyi Tian, Ting Wei.

**Formal analysis:** Celestin Hategeka, Margaret E. Kruk.

**Investigation:** Celestin Hategeka, Robert Marten, Ruitai Shao, Margaret E. Kruk.

**Methodology:** Celestin Hategeka, Robert Marten, Ruitai Shao, Margaret E. Kruk.

**Project administration:** Celestin Hategeka.

**Supervision:** Margaret E. Kruk.

**Visualization:** Celestin Hategeka, Margaret E. Kruk.

**Writing – original draft:** Celestin Hategeka, Margaret E. Kruk.

**Writing – review & editing:** Celestin Hategeka, Prince Adu, Allissa Desloge, Robert Marten, Maoyi Tian, Ting Wei, Margaret E. Kruk.

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
