## [Editor Report · Decision Letter 0]

10 Dec 2021

Dear Dr Hategeka, 

Thank you for submitting your manuscript entitled "Systematic Review of Implementation Research on Non-Communicable Disease Care in Low- and Middle-Income Countries" for consideration by PLOS Medicine.

Your manuscript has now been evaluated by the PLOS Medicine editorial staff and I am writing to let you know that we would like to send your submission out for external peer review.

Please re-submit your manuscript within two working days, i.e. by Dec 14 2021 11:59PM.

Kind regards,

Beryne Odeny

PLOS Medicine

---

## [Decision Letter · Decision Letter 1]

28 Jan 2022

Dear Dr. Hategeka,

Thank you very much for submitting your manuscript "Implementation Research on Non-Communicable Disease Care in Low- and Middle-Income Countries: A Systematic Review" (PMEDICINE-D-21-05057R1) for consideration at PLOS Medicine. 

Your paper was evaluated by a senior editor and discussed among all the editors here. It was also sent to independent reviewers, including a statistical reviewer. The reviews are appended at the bottom of this email and any accompanying reviewer attachments can be seen via the link below:

[LINK]

In light of these reviews, I am afraid that we will not be able to accept the manuscript for publication in the journal in its current form, but we would like to consider a revised version that addresses the reviewers' and editors' comments. Obviously we cannot make any decision about publication until we have seen the revised manuscript and your response, and we plan to seek re-review by one or more of the reviewers. 

We expect to receive your revised manuscript by Feb 18 2022 11:59PM. Please email us (plosmedicine@plos.org) if you have any questions or concerns.

We look forward to receiving your revised manuscript. 

Sincerely,

Beryne Odeny, 

PLOS Medicine

plosmedicine.org

1) Please add this statement to the manuscript's Competing Interests: "Margaret Kruk is an Academic Editor on PLOS Medicine's editorial board."

2) Abstract:

a) Please report your abstract according to PRISMA for abstracts, following the PLOS Medicine abstract structure (Background, Methods and Findings, Conclusions) http://www.plosmedicine.org/article/info:doi/10.1371/journal.pmed.1001419 .

b) Please combine the “Methods” and “Findings” sections into one “Methods and findings” section 

c) Please ensure that all numbers presented in the abstract are present and identical to numbers presented in the main manuscript text.

d) Please summarize the synthesis/appraisal methods

3) Please remove the “Research in Context” section

5) Please submit a completed PRISMA checklist. We understand that not all items on the checklist will be directly relevant, and these can be marked as ‘not applicable’. 

a) The PRISMA guidelines provided at the EQUATOR site http://www.equator-network.org/reporting-guidelines/prisma/

6) Please rename figure 1 to “PRISMA flow chart”

7) In line with PLOS Medicine’s guidelines, please update your search to the present time and provide the beginning and end dates of your search.

8) References: 

a) Please select the PLOS Medicine reference style in your citation manager. In-text reference call outs should be presented as follows noting the absence of spaces within the square brackets, e.g., "... countries [1,2]."

b) Please ensure that journal name abbreviations consistently match those found in the National Center for Biotechnology Information (NCBI) databases. https://journals.plos.org/plosmedicine/s/submission-guidelines#loc-references. 

c) Please ensure six names appear before et al. e.g., check refs #26-32, 34-36, 38 and so forth

Comments from the reviewers:

Reviewer #1: See attachment

Michael Dewey

Reviewer #2: Reviewer Comments

Title: Systematic Review of Implementation Research on Non-Communicable Disease Care

in Low- and Middle-Income Countries

Authors: Hategeka C, et al. 

Overview

The investigators conducted a systematic review of evidence on the current state of implementation research (IR) for NCD prevention and control in low-and middle-income countries (LMICs) using MEDLINE and EMBASE databases from 1990 through 2020. They used standard search terms for this topic that are detailed in their documentation. Among more than 9600 potential studies they found 222 eligible studies from 63 countries. Most studies were on cervical cancer and were for proof of concept or pilots targeted at a micro-level of the healthcare system. Most studies used quantitative methods and weak study designs were common. Study publications increase dramatically over their study timeframe. Major gaps in IR were noted and that published studies were mostly funded by international sources. 

General Comments

This is a very well written study that is very important and timely. IR is currently vastly underused in LMICs, and the authors highlight this challenge and the need for more efforts and resources. While they do show convincingly that the number of IR studies annually has increased over their 30-year study timeframe, efforts in this area fall short of what is needed. 

There are several areas where the manuscript could be improved. The results seem to be in both the results section some are also in the discussion section. There are many figures and tables and some of the findings could be moved into the text with a concise description and allowing for deletion of some figures/tables. Better description of the inclusion/exclusion criteria would be helpful. A brief clear description of the working definition of screening, treatment and prevention would be useful. 

Some discussion on why so few studies were multinational would be helpful. Also, a brief description of micro, meso and macro health system interventions would be useful. In the discussion, using high income country comparable statistics might be useful to give the context of what is currently the situation (e.g., use of implementation science framework for studies in HIC vs LMIC, etc.). While not the ideal comparator is could help under the challenges in the LMIC context. In the discussion/conclusion, it would be helpful to present some of the challenges and barriers conducting IR in LMICs. This could be very insightful. Finally, a better description of why descriptive epidemiological studies were excluded is needed. These may contain barriers and facilitators to implementation and be key in designing IR intervention strategies. 

Specific Comments

Funding source section.

While the statement is made that the funders have no involvement in the study, it does not state who the funders were. Suggest including them. 

Page 8 first para

…"country of implementation" is a bit unclear. Suggest making it "county where the IR is being conducted" - it seems that is what you are communicating. 

Figure 1. 

More description of the exclusion categories is needed. For example, intervention, and full text are not entirely clear. 

Figure 2

How were studies that had multiple interventions in this graphic? Were they included in all the categories that had interventions? 

Figure 3. 

This plot is driven by county size. Consider a metric that uses a standardization method such as studies per 100,000 population. This may allow for better comparisons, etc. 

Figure 5. 

Seems that A and B should have the same total numbers but in B 2020 is lower than 2020 in A. Please explain/reconcile.

Figure 7. 

Same issue as for Fig 2. How were studies with multiple interventions handled. Some studies may have a N >1. Please clarify. 

Table 1. 

See above comment about exclusion of epidemiology descriptive studies. They may have barriers and facilitates for IR intervention development. 

Appendix Table 3. 

Should have a date on it because LMIC countries change their status over time (e.g., move from LIC to LMIC, etc.)

Appendix Figure 1. 

Diabetes is misspelled

Appendix Figure 5. 

See above comments. Macro, Meso, and Micro need short descriptions.

Appendix Figure 7. 

The legend is off the page. 

Appendix Figure 8. 

Legend is confusing as a standalone. Suggest: Funding, no funding (at least none noted in the published report). However, for the study to happen, it must have been funded at some level - so 'no funding" may not work. More detail would be interesting. 

Reviewer #3: Thank you for this interesting and useful review of the literature focusing on implementation research on 33 WHO recommended interventions to prevent and control non-communicable diseases in low- and middle-income countries.

General comments

Overall, the study is well written and easy to follow. I have a series of more specific suggestions and comments below, but my main concern is that you only present descriptive statistics regarding some key characteristics of the identified studies, rather than the narrative synthesis which you planned to do according to the study protocol in PROSPERO. Such a narrative synthesis could have provided the reader with a richer understanding of what included studies found and other aspects beyond the narrower quantitative approach used. If you can, including a narrative synthesis would likely be very interesting. If this is not feasible, you should at least comment on this departure from the study protocol and throughout the paper more clearly state the narrow quantitative scope of the study.

Another key aspect which is missing from the paper is an assessment of the quality and usefulness of the included studies. Indeed, you stress the importance of good quality implementation research in several places in the manuscript, yet it is difficult from this study to say anything about the current quality of implementation research on NCD interventions in LMICs.

Specific comments

-- Title

Since the study focuses on NCD prevention and control interventions and not only care, you could consider replacing "Care" with "prevention and control interventions" in the manuscript title: "Implementation Research on Non-Communicable Disease Prevention and Control Interventions in Low- and Middle-Income Countries: A Systematic Review"

-- Abstract

I would not mention the protocol in the abstract.

"We synthesize extracted data narratively using descriptive statistics" - generally narrative synthesis refers to "the use of words and text to summarise and explain the findings of the synthesis", whereas descriptive statistics is a quantitative approach to describe or summarize the characteristics of a sample or data set. Actually, in this study you only present descriptive statistics, no narrative summary or synthesis. Please revise the abstract accordingly.

It would be useful if you could mention something about inclusion and exclusion criteria in methods.

"…approximately similar to their proportion of global population…" while this is a nice idea, it does not make sense to make this sort of comparison since you are missing the high-income countries in your review and they would very likely distort these proportions. Also, your assessment isn't quite correct since LICs are home to about 9% of the global population and about 73% live in MICs.

"Slightly more studies used the weakest study design, pre-experimental, than the strongest design, experimental (25% versus 24%)" - in my view, it would be more correct to state that these two categories were (almost exactly) equal as the difference is something like 2 studies out of 222.

"Despite growth in implementation research on NCDs in LMICs" - this statement is not supported by the findings in the abstract, suggest adding a line supporting this.

"Future studies should prioritize implementation at scale". One could argue that studying interventions at the pilot stage should be prioritized in order to assess whether to take an intervention to scale in the first place. Is there a risk that advocating for a priority on studying implementation at scale could lead to less well studied interventions going to scale?

"stronger internal validity, be more conceptually-driven and use mixed- methods to understand mechanisms" - while interesting, these conclusions are not supported by the findings stated in the abstract. Note that it should be possible to read the abstract without reading the full manuscript.

"To maximize impact of the research under limited resources, adding implementation science outcomes to effectiveness research and regional collaborations are promising." Could you clarify what you mean by this statement, and consider whether this conclusion is supported by the findings of the present study?

-- Research in context

The text provided under "Added value of this study" does not seem to correspond to this heading. I would suggest beginning the paragraph with something like "This study provides the first comprehensive review of implementation research on NCD interventions…"

The text provided under "Implications of all the available evidence" does not seem to be supported by the findings of the present study. Please review this paragraph.

-- Introduction

In the first paragraph, you only mention NCD mortality. I would encourage you to also mention the significant NCD morbidity and its consequences.

You mention COVID-19 and the interlinkages between NCD morbidity and COVID-19. You could consider also making a similar link to other infectious diseases such as TB and HIV to drive home the point that NCD prevention and control should not be regarded as a vertical issue separated from other health issues.

You write that "…the evidence for the clinical effectiveness of most NCD prevention and treatment interventions is well established…" - would you agree with Isaranuwatchai et al. (BMJ 2020;368:m141) who argue that the clinical effectiveness of some NCD prevention and treatment interventions may indeed be contestable or even wasted in some contexts?

Please consider explicitly stating the study aim and/or research question of the study.

Given the limited scope of this study, consider rephrasing "In this systematic review, we synthesize evidence on the current state of implementation research" to better reflect that you present descriptive statistics of key characteristics of peer-reviewed publications on implementation research on 33 specific interventions in LMICs in 1990-2020.

-- Methods

According to the information registered on the PROSPERO database, the study protocol was technically not pre-specified as the start date (Dec 1, 2020) was seven months before the protocol was registered (July 30, 2021), and the anticipated completion date was only one day later (July 31, 2021).

Could you clarify in the methods how you selected the included interventions and specifically how you define "relevant" interventions and "within health systems"? For example, in Table 2 / Appendix table 1 you list "mass media campaigns that educate the public about the harms of smoking/tobacco use and second hand smoke" as an intervention within the health system, whereas it seems like you have excluded other interventions such as taxation and plain packaging. 

"Low- and middle-income countries as defined by the World Bank." - please specify which version you use as the classification is modified each year and the list has changed a fair bit since 1990. For example, in 1990 both China and India were classified as low-income countries.

Throughout, I would recommend making the reference to the updated Appendix 3 of the WHO Global NCD Action Plan 2013-2020 more explicit to avoid the perception that you came up with your own list of interventions.

The list of data elements you extracted, presented in appendix table 4, plays a central role in defining the focus of your study. Please describe how you arrived at the 29 included parameters. 

The following sentence is redundant since the information is provided in table 1: "Non-empirical/primary research including reviews meta-analyses, editorials, commentaries, letters to editors, opinion papers, newspaper and protocols, are not eligible for inclusion."

As noted in the abstract: As far as I can see, you do not present any narrative summary or synthesis of findings, only some basic descriptive statistics.

The heading "Risk of bias assessment" could be removed from methods since it was not done, unless it is explicitly required by editors. The corresponding paragraph could be moved to the limitations section.

Table 1. This table is unnecessarily unwieldy. Specifically I would recommend removing the row for "population" as your review focuses on interventions and not populations (that non-humans are excluded can be left implicit in my opinion), and significantly shorten the study design section - it is not necessary to list all existing study designs here.

-- Results

Perhaps I am missing something, but I would normally expect there to be a table with the full list of included studies and their characteristics (either in manuscript or in appendix). Currently, there is just a reference to Table 1 (which should not be referenced here as it belongs to methods), followed by references "20-241". In my view, this is not a sufficient way to list the included studies.

I would recommend using the same number of decimal points throughout (you now mix between 1 and 0 decimal points).

You refer to a grand total of 12 figures in the Appendix, all of which are simple bar or pie charts. In my opinion, this information would be far more easily digested if you could present it as one or two tables instead.

According to table 2 you have only 33 eligible interventions in this study - how could you find 265 interventions? Per the study design, you should only be able to find a maximum of 33 interventions. Could you clarify this? (According to Appendix figure 1, you found studies covering 32 of the 33 eligible interventions, and in total there were 265 instances across the 32 interventions).

Appendix figure 1. The figure says "Fiabetes management", please review.

How do figure 2 and appendix figure 1 relate? The numbers seem not to match.

Appendix table 5 should have a column with country income category. The column entitled "proportion" should read "percentage".

Sorry to make an editorial comment, but please make sure the order of figures and tables are correct (for example, Appendix figure 2 appears in the text before Appendix figure 1).

You state that "The NCD conditions targeted varied by countries" and refer to Appendix Figure 3, but the figure only shows how the number of interventions targeting each condition (though you write number of conditions evaluated) varied by country income category. Please revise the text and figure.

Could you clarify what you mean by "The NCD interventions varied by conditions and type" - in what way did interventions vary? Perhaps you mean that the number of identified interventions varied?

When you state "The intervention focus appears to vary by countries" you again seem to be referring to the country income category rather than country.

The three types of intervention focus brought up in results (screening, treatment and prevention) do not appear to be presented neither in the methods nor in the appendix - where Appendix table 4, row 22. "intervention type" refers back to Appendix table 1, where in column "category of interventions" you only have primary, secondary or tertiary prevention. Please make sure that all methods used are presented in the methods and that the same terminology is used throughout.

Please include a definition and reference for what you mean by "level of health system" (micro, meso, macro) in the methods as this is not clear to the reader.

Each time you write that study characteristics varied by income category or condition etc. I wonder whether you did any simple statistical test to assess whether there were significant differences, or did you decide against this?

I would suggest mentioning the equity lens in the methods since you bring it up in the results (even though it is presented in Appendix table 4).

Figure 5A presents the number of publications by year, but it generally takes years from research to publication - do you have any sense of when research was carried out, not only when it was published?

-- Discussion

I am not sure I agree with this statement: "These studies therefore represent the state of the science today on how to scale up the response of the health system to the growing burden of NCDs in the countries bearing the bulk of disease burden from non-communicable conditions." - nowhere in your study design do you mention a focus on "how to scale up the response of the health system", or is this implicit somehow?

Please rephrase as discussed above: "The studies described 265 different NCD interventions, ranging from screening to prevention to treatment and palliation."

The comparison between number of studies and disease prevalence is interesting but I would recommend adding it to results (and methods) rather than introducing this analysis in the discussion.

Some of the statements in the discussion appear to be new information, i.e. "We found that only 31 of the studied interventions addressed prevention with nearly 80% of these tackling primary prevention (e.g., tobacco use, inactivity, unhealthy diet)." - could you please make sure that you do not introduce new information in the discussion?

You write: "arguably WHO- proposed interventions are well established and a range of implementation models abound" - please see my earlier comment about the fact that the list of interventions has been questioned in its own right. I also note that you contradict yourself when you write that "a range of implementation models abound", while in the introduction you noted that "care delivery models and means of scaling these up to entire populations in need in heterogeneous and resource-constrained health systems are not".

"the micro-level of the health system— targeting patient, provider, or clinic levels", this is the first time in tha manuscript that you explain what you mean by the micro level, see comment above.

"Nearly one in five tested a new technology, despite evidence that technology adoption without substantial integration into policies, data, and workflows is typically ineffective in transforming care" - are you here implying that this was not the case in the included studies?

"The lack of domestic support is unlikely to be offset by global funding going forward; a recent analysis showed that NCDs were under prioritized in bilateral agency portfolios relative to their health impacts." - I would argue that this situation is changing, besides, it may be more useful to think about this in terms of health systems strengthening than as a vertical NCD silo. You may want to touch on that aspect in the discussion.

You write "We had no language restrictions permitting a comprehensive assessment of the published literature" - did you do any searches in other languages? Did you include studies in other languages?

I would suggest adding a few sentences to the limitations about the fact that you did not assess quality or usefulness of the included studies. (Indeed, you begin the conclusion by stating that "High quality implementation science can play a key role in informing effective delivery of health system interventions", yet your study only quantifies the number of studies, not their quality.)

Limitations should also, importantly, discuss the fact that there may be significant implementation research conducted which is never published which may introduce important selection bias in your study.

The paragraph starting with "Over 50 of the 222 included studies used an experimental research design." seems misplaced after the "strength and limitations" section.

-- Conclusion

You should generally avoid adding new arguments and references in the conclusion, but rather conclude what you found, relating back to your study aim. Your current conclusion reads more like a conclusion about the usefulness and need for implementation more broadly, rather than about your study findings. Several of the conclusions are not supported by the study findings and should be brought up in the discussion rather than the conclusion. Please revise the conclusion accordingly.

[LINK]

---

## [Decision Letter · Decision Letter 2]

29 Mar 2022

Dear Dr. Hategeka,

Thank you very much for submitting your manuscript "Implementation Research on Non-Communicable Disease Prevention and Control Interventions in Low- and Middle-Income Countries: A Systematic Review" (PMEDICINE-D-21-05057R2) for consideration at PLOS Medicine. 

[LINK]

In light of these reviews, I am afraid that we will not be able to accept the manuscript for publication in the journal in its current form, but we would like to consider a revised version that addresses the reviewers' and editors' comments. Obviously we cannot make any decision about publication until we have seen the revised manuscript and your response, and we plan to seek re-review by one or more of the reviewers. 

We expect to receive your revised manuscript by Apr 19 2022 11:59PM. Please email us (plosmedicine@plos.org) if you have any questions or concerns.

We look forward to receiving your revised manuscript. 

Sincerely,

Beryne Odeny, 

PLOS Medicine

plosmedicine.org

Requests from the academic editor:

This is an interesting study, but I think two things are missing and could make it much better.

First, there is a notable lack of attention in systematically capturing categorizing the strategies. While the categorization of strategies is a moving target in implementation science, it is nevertheless a crucial "ingredient" in implementation science and our limited ability to describe the strategy (which is the exposure or treatment if you will in this field) limits what we learn from it. There are many different ways of classifying strategies, that range from the original Proctor Actor-Action etc., to ERIC to Cochrane EPOC to Waltz et al. - none of them are perfect but they can still be useful. At a minimum, who in the health system is carrying out the strategy, what are the actions, and who are the actions meant to target are critical. In truth, some of the strategies are going to be difficult to categorize because the primary sources do not say what they are, but it would nevertheless be of interest say how many did or did not. The authors are undoubtedly familiar with both the challenges and the importance of describing strategies used in these studies, so whatever direction they take, some discussion and justification of it in the discussion would be helpful.

Some additional analysis of the implementation outcomes might be useful, with more description on where in the health system (at the patient, the hcw, the organization, the system) etc., the strategies targeted. How has this changed over time? does it differ by disease condition? what about by donor? Likewise this could be done with the strategies as well, but understanding what kind of targets implementation science addressing NCD's seek would be helpful. What I would be concerned about in general is the phenomenon where clinical research and patient level outcomes in clinical medicine drive outcome selection and measurement, and even though much of the action is above the level of the patient, the outcomes - even those considered implementation outcomes - are reported at the level of the patient, and if so would be of limited informativeness.

Comments from the reviewers:

Reviewer #1: The authors have met most of my points especially about the figures although I still think Figure 4 is not really a good use of ink and space.

I thank the authors for the offprint about comparison of databases but I do not draw the same optimistic message from it as they do. Unfortunately it does not give us the breakdown of the intersections of the coverage so we cannot tell if searching a new database would give us additional references. Just because Embase has, say 10%, and AIM 5% does not imply the 5% are a proper subset of the 10%. They might all be different. I still think the search is rather limited.

Michael Dewey

Reviewer #3: Thank you for your thorough and thoughtful responses to reviewers' and editors' comments. As this is a re-review, I present my remaining concerns below.

You state that "We have already clarified in our methods section and in the protocol that narrative descriptive analysis that would be used in this review refers to descriptive statistics including summary statistics of type of interventions, study designs, and implementation outcomes". My point is that you cannot do descriptive statistics and call it narrative synthesis as those are two different things. Please revise the manuscript accordingly, possibly including an explanation why you did not do a narrative synthesis as you had planned to do according to the study protocol.

You state that "Table 2 shows a list of unique interventions that were eligible for inclusion. In the results, we report total number (instead of unique type of interventions) of NCD interventions evaluated in 222 included studies. We found that 265 interventions were evaluated across 222 included studies, meaning that there were some studies that evaluated more than one intervention.". I do not think you have addressed the problem I raised regarding the terminology used, namely that you use the word "intervention" to mean both the 33 priority NCD interventions included in this study and the 265 studied instances of those interventions being used. This is particularly confusing since you actually found 222 studies covering 32 of the 33 interventions. I suggest you consider an alternative terminology which separates between the 33 priority interventions and the 265 instances of interventions being used.

Regarding your comment that "We reported year of publication and not time of when study/implementation was conducted." I would suggest you make a comment about this in the limitations.

[LINK]

---

## [Editor Report · Decision Letter 3]

1 Jun 2022

Dear Dr. Hategeka,

Thank you very much for re-submitting your manuscript "Implementation Research on Non-Communicable Disease Prevention and Control Interventions in Low- and Middle-Income Countries: A Systematic Review" (PMEDICINE-D-21-05057R3) for review by PLOS Medicine.

I have discussed the paper with my colleagues and the academic editor. I am pleased to say that provided the remaining editorial and production issues are dealt with we are planning to accept the paper for publication in the journal.

[LINK]

We look forward to receiving the revised manuscript by Jun 08 2022 11:59PM.   

Sincerely,

Beryne Odeny, 

PLOS Medicine

plosmedicine.org

Requests from Editors:

1) Figure 3 - At the top of each column, please indicate the meaning of numbers in each cell/ row (e.g., “=5”). If this is the N of studies, please clearly indicate this at the top of each column.

2) References 

a) Please ensure that journal name abbreviations consistently match those found in the National Center for Biotechnology Information (NCBI) databases. https://journals.plos.org/plosmedicine/s/submission-guidelines#loc-references. 

b) Please ensure that an access date is provided for all references with weblinks.

Comments from Reviewers:

[LINK]

---

## [Editor Report · Decision Letter 4]

16 Jun 2022

Dear Dr. Hategeka,

Thank you very much for re-submitting your manuscript "Implementation Research on Non-Communicable Disease Prevention and Control Interventions in Low- and Middle-Income Countries: A Systematic Review" (PMEDICINE-D-21-05057R4) for review by PLOS Medicine.

I have discussed the paper with my colleagues and the academic editor. I am pleased to say that provided the remaining editorial and production issues are dealt with we are planning to accept the paper for publication in the journal.

[LINK]

We look forward to receiving the revised manuscript by Jun 23 2022 11:59PM.   

Sincerely,

Beryne Odeny, 

PLOS Medicine

plosmedicine.org

Requests from Editors:

- In the abstract, methods and discussion sections, please clearly mention that the implementation strategies are not specified well enough. This should be highlighted as one of the limitations.

[LINK]

---

## [Editor Report · Decision Letter 5]

21 Jun 2022

Dear Dr Hategeka, 

On behalf of my colleagues and the Academic Editor, Dr Elvin Hsing Geng, I am pleased to inform you that we have agreed to publish your manuscript "Implementation Research on Non-Communicable Disease Prevention and Control Interventions in Low- and Middle-Income Countries: A Systematic Review" (PMEDICINE-D-21-05057R5) in PLOS Medicine.

PRESS

Sincerely, 

Beryne Odeny 

PLOS Medicine